# Biologically Active Compounds in Selected Organic and Conventionally Produced Dried Fruits

**DOI:** 10.3390/foods9081005

**Published:** 2020-07-27

**Authors:** Dominika Średnicka-Tober, Renata Kazimierczak, Alicja Ponder, Ewelina Hallmann

**Affiliations:** Department of Functional and Organic Food, Institute of Human Nutrition Sciences, Warsaw University of Life Sciences, Nowoursynowska 159c, 02-776 Warsaw, Poland; dominika_srednicka_tober@sggw.edu.pl (D.Ś.-T.); renata_kazimierczak@sggw.edu.pl (R.K.); alicja_ponder@sggw.edu.pl (A.P.)

**Keywords:** dry fruits, organic, conventional, polyphenols, carotenoids, basket study

## Abstract

A growing demand for organic foods is driven by consumers′ perception that they are more nutritious. However, while there is a number of scientific studies showing some superior qualities of organically grown fruit and vegetables, including, i.e., higher content of phenolics, some vitamins and antioxidant activity, scientific research looking into the quality of processed organic products is very limited. At the same time the consumption of processed, convenient foods, with a long shelf-life, is rapidly increasing all over the world. Among the processed fruit-based products, dried fruits are recognized by consumers as one of the best snacks, highly nutritious and containing a large amount of fibre. In the presented experiment, four types of organic and conventionally produced dried fruits were examined: Apricots, apple rings, cranberries, and prunes. The concentrations of polyphenols (in all products) and carotenoids (in apricots) were measured using high performance liquid chromatography (HPLC). The study confirms that dry fruits can be undoubtedly considered as a rich source of polyphenols, however, a large variation in the concentrations of these compounds among different brands of products was also pointed: 219.03 ± 3.90–296.96 ± 2.86 mg 100 g^−1^ in dried apricots, 95.24 ± 15.12–627.71 ± 48.64 mg 100 g^−1^ in dried apple rings, 14.64 ± 0.43–203.09 ± 7.96 mg 100 g^−1^ in dried cranberries, and 134.65 ± 12.27–422.44 ± 9.00 mg 100 g^−1^ in prunes. Carotenoids concentrations in dried apricots ranged from 2.72 ± 0.31 to 17.49 ± 0.17 µg g^−1^. Among the tested dried fruits, only in the case of apricots organic products were characterized by the higher contents of phenolics and carotenoids compared to the conventional brands. In the case of other products there was either no consistent significant production system effect, or the concentrations of the analyzed phenolic compounds were higher in conventional products.

## 1. Introduction

Fruit consumption is one of the most important elements of a properly balanced diet [1]. Fruits contain many valuable nutrients and biologically active compounds such as, e.g., polyphenols, carotenoids, and vitamins with a positive effect on human health. Regular fruit consumption can protect the body against the development of numerous metabolic diseases associated with incorrect nutrition habits contributing to a reduction in the incidence of various types of cancer and decreasing the risk of cardiovascular disease [2,3,4,5]. The biologically active compounds present in the fruits reduce the level of LDL (low-density lipoprotein) cholesterol in blood and lower blood pressure [6].

Many fruits are seasonal and thus are not available throughout the entire year. One of the oldest methods of fruit preservation, significantly extending the shelf life of the fruits, and thus the availability to the consumer, is drying. Dried fruits are easier to distribute, store, and transport. The fruits can be dried in their whole shape (grapes, cranberries, apricots, and plums) or sliced (kiwifruit, apple, mango, and papaya) [7]. Even though dried fruits contain high levels of carbohydrates, including sugars [8,9], they are recognized by consumers as one of the best snacks.

Dried apricots, apples, cranberries, and prunes belong to the most popular dried fruits. Appropriate drying conditions of these fruits allow providing products of a high nutritional quality, rich in biologically active compounds. Dried apricots are known to be a rich source of carotenoids (i.e., β-carotene and lutein), polyphenols, and especially chlorogenic acid and phytoestrogens. Phenolic acids and flavanols are found in prunes. Dried cranberries are a rich source of flavonoids and phenolic acids, particularly p-hydroxybenzoic acid [10,11].

The largest producer of dried apricots is Turkey. Over 61% of the world′s production of these dried fruits come from this region. The second top producer is Iran (14%), followed by China (3%), Uzbekistan (2%), and the USA (1%). The largest producers of dried cranberries are the United States (60%), Canada (35%), and Chile (5%). In the case of prunes, the largest producers are the USA (43%), Chile (30%), France (14%), and Argentina (9%) [12].

Fruits in the organic system are produced without the use of synthetic fertilizers and pesticides, following the organic farming standards [13,14]. Organic practices are based on natural fertilizers, such as compost and manure, and diverse rotations. Pest protection takes the form of natural predators, pheromone traps, and other biological and mechanical methods [15]. Organic processing allows only natural, minimally invasive methods, and the use of synthetic food additives is prohibited. A growing demand for such naturally produced organic foods is strongly driven by consumers′ perception that they are more nutritious and health-beneficial. However, while there is a significant number of scientific studies showing a superior quality of organic raw materials, including, i.e., higher contents of phenolics, some vitamins and higher antioxidant activity [16], scientific research looking into the quality of the processed organic products is very limited. It is clearly an important challenge to bring the health-related attributes of organic fruit and vegetables from field to fork through the processing step.

In the presented study, we have examined the concentrations of selected groups of bioactive compounds, such as carotenoids and phenolics, in organic and conventionally produced dried apricots, apple rings, cranberries, and prunes available on the Polish market.

## 2. Materials and Methods

### 2.1. Chemicals

Acetone HPLC purity (Sigma-Aldrich, Warsaw, Poland); carotenoids standards (99.9% purity): Lutein, zeaxanthin, α-carotene, β-carotene (Sigma-Aldrich, Warsaw, Poland); magnesium carbonate (Chempur, Warsaw, Poland); methanol HPLC/MC purity (Sigma-Aldrich, Warsaw, Poland), *ortho*-phosphoric acid 85% (Chempur, Warsaw, Poland); polyphenols standards: Gallic acid, chlorogenic acid, caffeic acid, benzoic acid, ferulic acid, *p*-coumaric acid, *p*-hydroxybenzoic acid, kaempferol, kaempferol-3-*O*-glucoside, myricetin, quercetin, quercetin-3-*O*-rutinoside (Merck, Warsaw, Poland); ultrapure water (Sigma-Aldrich, Warsaw, Poland).

### 2.2. Equipment

Food processor (Kenwood FDM 790 BA, Poland), laboratory scale WPC 510/S/2 (RadWag, Warsaw, Poland), hot-dryer (FP-25W Farma Play, Warsaw, Poland), HPLC-set: Shimadzu equipment (two LC-20AD pumps, CMB-20A system controller, SIL-20AC autosampler, UV-Vis SPD-20AV detector, Shimadzu, USA Manufacturing Inc., Canby, OR, USA).

### 2.3. Samples

Samples of conventional and certified organic dried apricots, apples, cranberries, and prunes were bought in the stores in Warsaw. In the case of each commodity, products of six popular brands were selected (three or four organic and two or three conventional brands, depending on the product). For each of the four tested products, brand names were coded with numbers (1–6). These number codes were consequently used in the results′ tables. Every sample was composed of six fruit bags (250 g each). Dried fruit samples were mashed in a mechanical food processor (Kenwood FDM 790 BA, Poland) to a pulp. The pulp was then used for dry matter, carotenoids, and polyphenols analyses.

### 2.4. Dry Matter Analysis

The total dry matter content of each dried fruit sample was measured by a gravimetric method. First, the mass of an empty glass vessel was measured using a laboratory scale. Next, one gram of fruit pulp was put into the vessel, and the mass was measured again. After 36 h of drying at 105 °C, the samples were cooled to room temperature and weighed for a third time. The total dry matter content in the examined samples was then calculated. The results are expressed as g 100 g^−1^ of the product [17].

### 2.5. Carotenoids Extraction and Identification

The carotenoids content in the dried apricots was measured using High Performance Liquid Chromatography (HPLC), following the method described by Nishiyama et al. [18] with modifications. A 100 mg sample of a dried-fruit pulp was mixed by a Vortex mixer with 5 mL of pure acetone. Next, samples were incubated in an ultrasonic bath (15 min, 0 °C) and centrifuged (6000 rpm, 10 min, 0 °C). The supernatant was transferred to dark HPLC vials. A Max-RP 80A column (250 × 4.6 mm) was used for the compounds separation. The analysis was carried out with a wavelength of 445–450 nm. Time of the analysis was 18 min. The injection volume was 100 μL. The chromatographic peaks corresponding to particular carotenoids were identified by comparing the retention times with those of authentic standards (Fluka and Sigma-Aldrich, Poznań, Poland, purity of 99.98%). This methodology was previously described by Kopczyńska et al. [19].

### 2.6. Phenolics Extraction and Identification

Concentrations of phenolic compounds in dried apple rings, apricots, cranberries, and prunes were determined by High Performance Liquid Chromatography (HPLC), as previously described by Hallmann et al. [20]. One hundred milligrams of dried fruit pulp was used to prepare the extract from every sample. The material was mixed with 5 mL of 80% methanol (*v*/*v* aqueous solution) and shaken in a Micro-Shaker 326 M (Poland). Then, it was placed in an ultrasonic bath (10 min, 30 °C, 5500 Hz). The sample was centrifuged (10 min, 3780 g, 5 °C). The decanted supernatant was re-centrifuged (5 min, 31,180× *g*, 0 °C), collected into vials and analyzed. The injection volume was 100 μL. For polyphenol compounds separation and identification, a Synergi Fusion-RP 80i Phenomenex column (250 × 4.60 mm) was used. The phenolic compounds were separated under gradient conditions with a flow rate of 1 mL min^−1^. Two gradient phases were used, 10% (*v*/*v*) acetonitrile and ultra-pure water (phase A) and 55% (*v*/*v*) acetonitrile and ultrapure water (phase B). The phases were acidified by ortho-phosphoric acid (pH 3.0). The wavelength used for detection was 270–360 nm. For compounds identification, the external standards of polyphenols with purities of 95.00–99.99% were used. The concentrations of polyphenols were calculated based on standard curve and sample dilution coefficients. This methodology was previously described in the study of Kopczyńska et al. [19].

### 2.7. Statistical Analysis

The obtained results were analyzed statistically in Statgraphics^®^ Plus 5.1 (Statgraphics Technologies, Inc., The Plains, VA, USA). Before the analyses, the normality of data distribution was verified using the qqnorm test. The results for three brands and two production systems were compared using the analysis of variance (ANOVA) followed by Tukey’s HSD post hoc test. The sample size was *n* = 6 for each brand, and *n* = 18 for each production system. Results are presented in tables, as means ± standard deviations. Values in the same row of the table followed by different letters are significantly different at the 5% level of probability.

## 3. Results and Discussion

### 3.1. Dried Apricots

The results obtained from the chemical analysis of the dried apricots are presented in Table 1. The brand had a significant effect on the concentrations of all bioactive compounds tested. The carotenoids concentrations in the tested samples ranged from 2.72 ± 0.31 to 17.49 ± 0.17 µg g^−1^ fw. The diversity of the results among the brands could have been expected, due to different drying methods, different sources of fresh material for dried apricot production (varieties not specified) or storage time.

At the same time significant impact of the production system (organic vs. conventional) on the carotenoids content (*p* < 0.0001), including α-carotene (*p* < 0.0001) and β-carotene (*p* < 0.0001), was observed (Table 1). Organically produced dried apricots contained significantly greater carotenoids content (*p* < 0.0001) compared to conventionally produced apricots, and this was consistent among all the tested brands. The higher total carotenoid content in organically produced dried apricots was unexpected. In organic processing a sulphur dioxide pretreatment is not allowed. At the same time this compound is widely used in conventional processing of apricots to prevent product spoilage and to retard the enzymatic and non-enzymatic browning reactions. While sulphur dioxide gives conventionally dried apricots their natural reddish-yellow colour, organically dried apricots are always deep brown after the drying process. Salur-Can et al. [21] determined the effects of sulphur dioxide on β-carotene in dried apricots during storage. The highest organic acid and β-carotene stabilities were found in sulphur-dried apricots.

The total polyphenols concentrations in the tested samples of dried apricots ranged from 219.03 ± 3.90 to 296.96 ± 2.86 mg 100 g^−1^. Organic dried apricots were significantly richer in polyphenols (total) compared to the conventionally produced ones (*p* < 0.0001), which was again consistent for all tested brands, and was mainly due to the high content of two major phenolic acids—gallic and chlorogenic acid—as well as higher concentration of the major flavonoid present in apricot fruits—quercetin-3-*O*-rutinoside (Table 1). This result is in line with the studies of other authors on organic fruits and vegetables composition, summarized in a large meta-analysis by Barański et al. [16], showing that organic crops are on average up to 70% richer in phenolic compounds when compared with the conventionally produced fruit. The authors, however, do not underestimate the importance of other factors such as, i.e., plant cultivar, climate, weather, specific agronomic practices, and finally also storage and processing techniques for the fruit quality, including concentrations of these health-promoting plant secondary metabolites. During storage, organic apricots can be more susceptible to rapid darkening and rapid decomposition of phenolic compounds than conventional apricots. A study conducted by Kan and Bostan [22] examined differences in the polyphenols and vitamin A content of organic and conventional fresh and dried apricots of different cultivars. The polyphenols and vitamin A contents in organically grown fruit were found to be higher than in those from conventional cultivation, in the case of all tested samples. These results are in line with the present study. The type of fruit drying method can also have a large impact on the content of phenolic compounds in dried apricots. Igual et al. [23] compared the drying kinetics and the change in the organic acids content, phenolics content, and antioxidant activity of dried apricots when using hot air drying and microwave energy. From the obtained results, they concluded that the industrial processing of dried apricots may be improved by using microwave energy, as the drying time was considerably reduced, and the obtained fruit had a higher phenolic content, particularly chlorogenic acid, catechin, and epicatechin.

### 3.2. Dried Apple Rings

Similarly to the apricots, the profile of phenolics in dried apple rings was strongly dependent on the brand. The concentrations of polyphenols (total) in the tested samples ranged from 95.24 ± 15.12 to 627.71 ± 48.64 mg 100 g^−1^ fw. This large among-brands variation was observed especially in the case of the major phenolic acid present in apples—chlorogenic acid—showing very low concentrations in apple rings of one of the two tested conventional brands. At the same time no consistent, significant effect of the production system (organic vs. conventional) on most of the analyzed phenolics was observed. The effects of the production system were identified only in the concentration of flavonoids (total) (*p* = 0.0001) and the major flavonoid—quercetin-3-*O*-rutinoside (*p* < 0.0001) (Table 2), with significantly higher concentrations of these compounds in conventionally produced products. The lack of identification of significant differences in the content of bioactive compounds among products from the two production systems was due to the large diversity of the phenolic compounds among the brands (Table 2). It could also have been affected by different technologies and apple cultivars used by each company to produce their dry apple rings (which was not controlled in the study). Apple cultivars have different susceptibilities to flesh darkening. Only non-darkening or less-darkening apple cultivars should be used for dry apple ring production [24]. Conventional producers use L-ascorbic acid as an antioxidant to prevent polyphenol oxidase activity and subsequently prevent the darkening of the apple flesh [25]. In organic processing, such practices are not allowed [13].

### 3.3. Dried Cranberries

Dried cranberry fruits appear to be a good source of phenolic compounds, especially phenolic acids [26], which has been confirmed in the present study. However, we have found large among-brands variation in the phenolics concentrations in dried cranberries (*p* < 0.0001 for all compounds tested) (Table 3). The polyphenols (total) concentrations in the tested samples ranged from 14.64 ± 0.43 to 203.09 ± 7.96 mg 100 g^−1^. The statistically significant effect of the production system, consistent among all tested brands, was identified only in the case of benzoic acid, which showed significantly higher concentrations in conventional products. It was expected that conventional dried cranberries could contain more phenolic compounds due to artificial benzoic acid and sodium benzoate being widely used as preservatives in conventional production, to ensure biological safety of products. These are actions taken to prevent the development of moulds and mycotoxins in the products during storage [27]. The study of Jeszka-Skowron et al. [28] indicated that using sodium benzoate may also inhibit the decomposition of phenolic compounds in dried fruits. In their study, Adamczak et al. [1] analyzed the influence of freeze and thermal drying on the content of organic acids and flavonoids in cranberry fruits. The results showed that the drying conditions have a significant effect on the content of bioactive compounds in cranberry fruits. In lyophilized fruit, distinctly more organic acids but fewer flavonoids were found than in fruit dried at a temperature of 35–40 °C. The mean flavonoid content in the thermally dried fruit was larger by 34 mg 100 g^−1^ compared to freeze-dried samples. Dorofejeva et al. [29] focused on the study of physical, chemical, and microbiological parameters changes in cranberries during convective drying. The results of the study showed that the content of polyphenols such as gallic acid, caffeic acid, and epicatechin decreased by 9.70, 9.90, and 11.68 mg, respectively, during treatment at temperatures up to + 50 ± 1 °C compared to the initial content of these compounds in non-dried cranberries.

### 3.4. Prunes

In the case of prunes, we observed highly significant differences among the brands, but also a strong effect of the production system, on the concentrations of the analyzed phenolic compounds (Table 4). The polyphenols (total) concentrations in the tested samples ranged from 134.65 ± 12.27 to 422.44 ± 9.00 mg 100 g^−1^. Phenolic acids, and especially gallic and chlorogenic acid, were the dominating phenolics in the prunes, next to much less abundant flavonoids. Conventional products were generally richer in polyphenols (total, phenolic acids, flavonoids) than organic ones, which in the case of most of the analyzed parameters was consistent among all the tested brands. Benzoic acid was found only in conventional samples. This finding may indicate the use of this compound as a preservative in conventional fruit processing. This creates a similar situation as in the case of dried cranberries, with inhibiting decomposition of the phenolic compounds in conventional dried products [30]. Vangdal et al. [31] analyzed the effect of the drying techniques on the retention of phytochemicals in conventionally and organically processed plums. The results of this study showed that the anthocyanin and ascorbic acid contents of prunes were significantly affected by the drying methods and that the cultivation system influenced the content of these compounds in prunes.

## 4. Conclusions

This study confirms that dry fruits can be undoubtedly considered as a rich source of polyphenols. However, a large variation in the concentrations of these compounds among different brands of products was also pointed: 219.03 ± 3.90–296.96 ± 2.86 mg 100 g^−1^ in dried apricots, 95.24 ± 15.12–627.71 ± 48.64 mg 100 g^−1^ in dried apple rings, 14.64 ± 0.43−203.09 ± 7.96 mg 100 g^−1^ in dried cranberries, and 134.65 ± 12.27−422.44 ± 9.00 mg 100 g^−1^ in prunes. Carotenoids concentrations in dried apricots ranged from 2.72 ± 0.31 to 17.49 ± 0.17 µg g^−1^.

Among the products tested within the presented study, only in the case of dried apricots organic products were characterized by the higher contents of phenolics and carotenoids compared to the conventional brands. In the case of the other products (dried apple rings, cranberries, and prunes) there was either no consistent, significant production system effect, or the concentrations of the analyzed phenolic compounds were higher in conventional compared to the organic products. While there is a growing evidence of superior quality of fresh organic fruits, including higher contents of phenolic compounds, some vitamins and higher antioxidant activity, it is clearly a challenge to bring these health-related attributes from field to fork through the processing step in the case of processed foods, such as dried products.

## Figures and Tables

**Table 1 foods-09-01005-t001:** The content of dry matter, carotenoids, and phenolic compounds in dried apricots of selected organic and conventional brands.

	Organic Brands			Conventional Brands			Organic	Conventional	*p*-Value
Compounds	1	2	3	4	5	6	all brands	all brands	Prod. system
Dry matter (g 100 g^−1^ fw)	80.27 ± 1.20 e ^1^	77.20 ± 0.85 c	75.61 ± 1.20 b,c	75.82 ± 0.13 b	79.25 ± 0.87 d	65.15 ± 1.26 a	77.69 ± 2.26 A	73.40 ± 6.41 A	N.S.^2^
Carotenoids (total) (µg g^−1^ fw)	17.49 ± 0.17 e	12.11 ± 0.04 d	9.34 ± 0.35 c	6.07 ± 0.33 b	2.72 ± 0.31 a	4.33 ± 0.24 b	12.98 ± 3.59 B	4.37 ± 1.47 A	<0.0001
Lutein	0.39 ± 0.01 c	0.27 ± 0.02 b	0.27 ± 0.03 b	0.25 ± 0.05 b	0.01 ± 0.01 a	0.36 ± 0.01 b	0.31 ± 0.06 A	0.21 ± 0.16 A	N.S.
Zeaxanthin	0.13 ± 0.03 c,d	0.14 ± 0.02 d	0.12 ± 0.01 b	0.09 ± 0.02 b	0.02 ± 0.01 a	0.36 ± 0.03 b,c	0.13 ± 0.02 A	0.16 ± 0.15 A	N.S.
α-carotene	0.55 ± 0.01 d	0.38 ± 0.02 c	0.34 ± 0.02 c	0.11 ± 0.01 b	0.07 ± 0.00 a	0.15 ± 0.01 b	0.42 ± 0.10 B	0.11 ± 0.04 A	<0.0001
β-carotene	16.41 ± 0.15 f	11.33 ± 0.07 e	8.62 ± 0.36 d	5.62 ± 0.26 c	2.61 ± 0.30 a	3.45 ± 0.27 b	12.12 ± 3.43 B	3.90 ± 1.36 A	<0.0001
Polyphenols (total) (mg 100 g^−1^ fw)	276.75 ± 1.74 c	278.27 ± 1.10 c	296.96 ± 2.86 d	239.57 ± 3.92 b	219.03 ± 3.90 a	234.52 ± 3.20 b	283.99 ± 9.91 B	231.04 ± 9.80 A	0.0001
Phenolic acids	132.12 ± 0.59 b	125.15 ± 1.21 a	154.43 ± 1.54 c	124.78 ± 1.15 a	128.21 ± 1.51 a,b	131.48 ± 2.96 b	137.23 ± 13.28 A	128.15 ± 3.39 A	N.S.
Gallic acid	62.55 ± 1.28 c	61.51 ± 0.76 b,c	86.89 ± 2.54 d	60.89 ± 0.34 b,c	57.74 ± 1.14 a,b	55.94 ± 1.75 a	70.32 ± 12.53 B	58.19 ± 2.41 A	0.0116
Chlorogenic acid	27.43 ± 0.17 f	26.09 ± 0.26 e	24.57 ± 0.14 d	20.83 ± 0.54 a	23.61 ± 0.24 c	22.53 ± 0.10 b	26.03 ± 1.25 B	22.32 ± 1.25 A	<0.0001
Caffeic acid	24.12 ± 0.44 a	23.03 ± 0.15 a	27.12 ± 0.59 b	28.43 ± 0.58 b	32.62 ± 0.23 c	34.71 ± 0.86 d	24.75 ± 1.87 A	31.92 ± 2.82 B	<0.0001
*p*-Coumaric acid	18.02 ± 0.31 c	14.53 ± 0.38 a	15.85 ± 0.39 b	14.63 ± 0.26 a	14.24 ± 0.08 a	18.30 ± 0.45 c	16.13 ± 1.56 A	15.72 ± 1.96 A	N.S.
Flavonoids	144.63 ± 2.31 d	153.11 ± 1.69 e	142.53 ± 1.76 d	114.79 ± 2.83 c	90.83 ± 2.39 a	103.04 ± 2.17 b	146.76 ± 5.13 B	102.89 ± 10.60 A	<0.0001
Quercetin-3-*O*-rutinoside	117.42 ± 1.72 e	123.76 ± 1.57 f	110.11 ± 1.91 d	85.69 ± 2.80 c	56.79 ± 1.54 a	66.19 ± 1.97 b	117.10 ± 6.10 B	69.56 ± 12.90 A	<0.0001
Myricetin	1.63 ± 0.03 a	1.75 ± 0.03 a,b	1.75 ± 0.42 a,b	2.10 ± 0.14 a,b,c	2.26 ± 0.06 b,c	2.38 ± 0.13 c	1.71 ± 0.22 A	2.25 ± 0.16B	<0.0001
Quercetin	23.05 ± 0.53 a	24.77 ± 0.96 a	27.52 ± 0.99 b	23.70 ± 0.08 a	27.41 ± 0.87 b	29.81 ± 0.02 c	25.11 ± 2.09 A	26.97 ± 2.70 A	N.S.
Kaempferol	2.53 ± 0.10 a	2.84 ± 0.07 b	3.15 ± 0.10 c	3.30 ± 0.06 c	4.36 ± 0.07 d	4.65 ± 0.10 e	2.84 ± 0.28 A	4.11 ± 0.62 B	<0.0001

^1^ Values in the same row followed by different letters (a–f; A–B) are significantly different at the 5% level of probability (*p* < 0.05); ^2^ not significant (N.S.).

**Table 2 foods-09-01005-t002:** The content of dry matter and phenolic compounds in dry apple rings of selected organic and conventional brands.

	Organic Brands				Conventional Brands		Organic	Conventional		*p*-Value
Compounds	1	2	3	4	5	6	all brands		all brands	Prod. system
Dry matter (g 100 g^−1^ fw)	93.90 ± 0.71 c ^1^	85.78 ± 0.31 b	91.72 ± 0. 52 c	81.52 ± 0.30 a	92.75 ± 0.32 c	92.18 ± 2.03 c	88.23 ± 4.91 A		92.46 ± 1.48 A	N.S.^2^
Polyphenols (total) (mg 100 g^−1^ fw)	118.88 ± 5.98 a	280.52 ± 1.49 b	393.65 ± 22.30 c	239.51 ± 4.83 b	95.24 ± 15.12 a	627.71 ± 48.64 d	258.14 ± 58.95 A		361.48 ± 68.66 A	N.S.
Phenolic acids	109.89 ± 6.44 b	264.51 ± 1.96 c	377.60 ± 22.39 d	213.90 ± 5.73 c	13.27 ± 2.03 a	590.37 ± 46.02 e	241.47 ± 57.10 A		301.82 ± 90.39 A	N.S.
Gallic acid	27.39 ± 0.96 c	2.98 ± 0.02 b	0.55 ± 0.05 a	0.10 ± 0.03 a	0.47 ± 0.12 a	0.48 ± 0.13 a	7.75 ± 11.40 A		0.47 ± 0.12 A	N.S.
Chlorogenic acid	67.19 ± 6.89 a	254.88 ± 1.31 b	352.17 ± 21.79 c	201.74 ± 6.05 b	3.62 ± 0.47 a	583.29 ± 45.04 d	218.99 ± 53.60 A		293.46 ± 91.58 A	N.S.
Caffeic acid	7.71 ± 0.24 c	3.94 ± 0.63 b	18.81 ± 0.79 d	3.77 ± 0.29 b	4.30 ± 0.46 b	2.01 ± 0.21 a	8.56 ± 6.15 A		3.16 ± 1.20 A	N.S.
Ferulic acid	7.60 ± 0.46 c	2.71 ± 0.10 a	6.06 ± 0.29 b,c	8.29 ± 0.12 c	4.87 ± 1.13 a,b	4.59 ± 1.15 a,b	6.17 ± 2.17 A		4.73 ± 1.15 A	N.S.
Flavonoids	8.99 ± 0.47 a	16.01 ± 0.56 a,b	16.05 ± 0.68 a,b	25.62 ± 1.46 a,b	81.98 ± 15.41c	37.33 ± 2.78 b	16.67 ± 5.98 A		59.65 ± 24.92 B	0.0001
Quercetin-3-*O*-rutinoside	5.97 ± 0.62 a	4.25 ± 0.29 a	2.27 ± 0.18 a	18.57 ± 1.15 a,b	63.78 ± 14.45 c	28.32 ± 2.28 b	7.77 ± 6.41 A		46.05 ± 20.53 B	<0.0001
Kaempferol-3-*O*-glucoside	0.82 ± 0.02 a	10.75 ± 0.41 c	9.31 ± 0.66 c	6.13 ± 0.83 b	17.37 ± 0.97 d	4.43 ± 1.22 b	6.75 ± 1.85 A		10.90 ± 3.56 A	N.S.
Quercetin	2.19 ± 0.13 a	1.01 ± 0.09 a	4.48 ± 0.06 b	0.93 ± 0.04 a	0.83 ± 0.03 a	4.59 ± 1.40 b	2.15 ± 1.44 A		2.71 ± 2.13 A	N.S.

^1^ Values in the same row followed by different letters (a–d; A–B) are significantly different at the 5% level of probability (*p* < 0.05); ^2^ not significant (N.S.)

**Table 3 foods-09-01005-t003:** The content of dry matter and phenolic compounds in dried cranberry fruits of selected organic and conventional brands.

	Organic Brands				Conventional Brands		Organic	Conventional	*p*-Value
Compounds	1	2	3	4	5	6	all brands	all brands	Prod. system
Dry matter (g 100 g^−1^ fw)	84.64 ± 0.13 b ^1^	80.69 ± 0.41 a	84.21 ± 0.49 b	84.66 ± 0.31 b	85.45 ± 0.40 b	84.79 ± 0.19 b	83.18 ± 0.63 A	84.97 ± 0.21 B	0.0218
Polyphenols (total) (mg 100 g^−1^ fw)	14.64 ± 0.43 a	201.88 ± 8.77 c	36.38 ± 1.81 a	81.31 ± 1.85 b	188.38 ± 10.97 c	203.09 ± 7.96 c	84.30 ± 28.03 A	157.59 ± 18.66 A	N.S.^2^
Phenolic acids	12.61 ± 0.41 a	200.47 ± 8.68 c	34.99 ± 1.81 a	80.28 ± 1.87 b	185.47 ± 11.06 c	201.15 ± 7.86 c	82.69 ± 28.08 A	155.64 ± 18.46 A	N.S.
Chlorogenic acid	0.39 ± 0.00 a	186.09 ± 7.96 c	29.77 ± 1.75 b	46.00 ± 1.03 b	173.60 ± 11.12 c	184.56 ± 7.09 c	72.08 ± 27.30 A	134.72 ± 21.42 A	N.S.
*p*-Hydroxybenzoic acid	1.95 ± 0.01 c	1.81 ± 0.08 c	0.16 ± 0.01 a	1.41 ± 0.10 b	0.28 ± 0.01 a	1.44 ± 0.12 b	1.31 ± 0.27 A	1.04 ± 0.19 A	N.S.
*p*-Coumaric acid	6.93 ± 0.46 c	11.41 ± 0.85 d	3.84 ± 0.08 b	14.88 ± 0.26 e	1.42 ± 0.03 a	3.07 ± 0.23 b	7.39 ± 1.08 A	6.46 ± 2.00 A	N.S.
Benzoic acid	3.34 ± 0.06 a	1.16 ± 0.14 a	1.21 ± 0.02 a	18.00 ± 1.41 c	10.17 ± 0.42 b	12.08 ± 1.07 b	1.91 ± 0.34 A	13.42 ± 1.26 B	0.0001
Flavonoids	2.03 ± 0.14 b	1.40 ± 0.09 a	1.39 ± 0.01 a	1.03 ± 0.03 a	2.91 ± 0.21 c	1.94 ± 0.13 b	1.61 ± 0.11 A	1.96 ± 0.27 A	N.S.
Kaempferol-3-*O*-glucoside	1.37 ± 0.10 c,d	0.93 ± 0.10 b,c	0.62 ± 0.02 a,b	0.39 ± 0.02 a	2.45 ± 0.21 e	1.41 ± 0.14 d	0.97 ± 0.11 A	1.42 ± 0.29 A	N.S.
Quercetin	0.66 ± 0.05 b	0.47 ± 0.01 a	0.77 ± 0.01 c	0.63 ± 0.02 b	0.45 ± 0.01 c	0.53 ± 0.01 a	0.63 ± 0.04 A	0.54 ± 0.03 A	N.S.

^1^ Values in the same row followed by different letters (a–e; A–B) are significantly different at the 5% level of probability (*p* < 0.05); ^2^ not significant (N.S.).

**Table 4 foods-09-01005-t004:** The content of dry matter and selected phenolic compounds in prunes of organic and conventional brands available on the Polish market.

	Organic Brands			Conventional Brands			Organic	Conventional	*p*-Value
Compounds	1	2	3	4	5	6	all brands	all brands	Prod. system
Dry matter (g 100 g^−1^ fw)	75.50 ± 0.38 c ^1^	75.94 ± 0.24 c,d	74.34 ± 0.24 c	67.25 ± 0.14 b	67.46 ± 0.11 b	61.39 ± 1.96 a	75.26 ± 0.76 B	65.37 ± 3.14 A	<0.0001
Polyphenols (total) (mg 100 g^−1^ fw)	134.65 ± 12.27 a	165.01 ± 16.26 a,b	215.22 ± 11.14 b,c	290.06 ± 24.13 d	254.54 ± 46.16 c,d	422.44 ± 9.00 e	171.63 ± 37.10 A	322.35 ± 81.06 B	0.0001
Phenolic acids	131.59 ± 12.02 a	163.19 ± 16.27 a,b	213.33 ± 11.15 b,c	284.76 ± 23.94 d	247.81 ± 46.31 c,d	415.22 ± 8.54 e	169.37 ± 37.52 A	315.93 ± 80.62 B	0.0001
Gallic acid	54.22 ± 1.93 a	77.37 ± 10.98 a,b	100.95 ± 9.33 b	74.35 ± 16.23 a,b	103.42 ± 20.67 b	106.19 ± 8.09 b	77.51 ± 21.50 A	94.66 ± 20.55 A	N.S.^2^
Chlorogenic acid	71.14 ± 12.27 a	81.98 ± 5.79 a,b	106.88 ± 3.34 b,c,d	93.81 ± 2.83 a,b,c	135.55 ± 27.27 d	122.08 ± 6.86 c,d	86.66 ± 17.34 A	117.15 ± 23.23 B	0.0061
*p*-Coumaric acid	5.09 ± 1.12 a	2.74 ± 2.84 a	4.35 ± 2.09 a	0.92 ± 0.66 a	7.14 ± 3.38 a	3.12 ± 1.48 a	4.06 ± 2.12 A	3.72 ± 3.31 A	N.S.
Ferulic acid	1.14 ± 0.01 a	0.051.10 ± 0.01 a	1.16 ± 0.01 a	1.03 ± 0.02 a	1.48 ± 0.14b	1.03 ± 0.02 a	1.13 ± 0.04 A	1.18 ± 0.23 A	N.S.
Benzoic acid	n.d.^3^	n.d.	n.d.	114.65 ± 14.67b	0.23 ± 0.23a	182.80 ± 7.01c	n.d.	99.23 ± 8.31	0.0019
Flavonoids	3.06 ± 0.26 b	1.82 ± 0.09 a	1.89 ± 0.02 a	5.30 ± 0.32 c	6.73 ± 0.16 d	7.21 ± 0.55 d	2.26 ± 0.62 A	6.42 ± 0.92 B	<0.0001
Kaempferol-3-*O*-glucoside	1.78 ± 0.28 b	0.62 ± 0.09 a	0.50 ± 0.03 a	0.68 ± 0.15 a	5.19 ± 0.16 c	2.01 ± 0. 24 b	0.97 ± 0.16 A	2.62 ± 0.1 B	0.032
Myricetin	0.90 ± 0.02 a	0.87 ± 0.01 a	0.92 ± 0.02 a	4.24 ± 0.22 b	1.20 ± 0.00 b	4.86 ± 0.38 c	0.90 ± 0.03 A	3.43 ± 1.71 B	0.0004
Quercetin	0.38 ± 0.009 b	0.34 ± 0.003 a	0.46 ± 0.010 c	0.38 ± 0.012 b	0.35 ± 0.001 a	0.35 ± 0.005 a	0.40 ± 0.055 A	0.36 ± 0.019 A	N.S.

^1^ Values in the same row followed by different letters (a–e; A–B) are significantly different at the 5% level of probability (*p* < 0.05); ^2^ not significant (N.S.); ^3^ not detected (n.d.).

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
