# Peer review of "Biologically Active Compounds in Selected Organic and Conventionally Produced Dried Fruits"

_foods, 2020, doi:10.3390/foods9081005_

Round 1

Reviewer 1 Report

The paper reports the identification of bioactive compounds in several organically and conventionally processed dried fruits. The study is useful in providing information about the beneficial compounds in the studied samples. However, the author is suggested to make some improvements to increase the quality of the manuscript.

Some irrelevant discussions appear in both the introduction and the discussion sections. The sentence construction and English grammar are also needed considerable help of improvements.

Abstract and conclusion:

- The report covered not only qualitative but also quantitative study. So, it would be better to provide both data in the abstract and conclusion.

- 24-32: These sentences are not related to the actual study; it is better to remove it from the abstract.

Introduction:

- The storyline is still confusing—no redlining between some sentences nor some paragraphs. The information provided by the author jumps from one description into another.

- 38: bioactive compounds is a very broad term. Vitamin, phenolic, and polyphenol are some examples of bioactive compounds. However, phenolics are not vitamin.

- Because the study only used several kinds of fruits, the justification for the selected fruit used in the study is needed.

- Additionally, a more precise judgment for the chosen analytes is also appreciated.

M&M:

- It would be better to separate the section for reagents and instruments used in the study. In this way, the reader could follow the information better.

R&D:

- Some irrelevant literature were discussed in this section.

- Discussing all the results into one big section is quite confusing. It is better if the author could arrange this section into several sub-sections and discuss the main idea for each topic separately.

- The identification result needs to be shown before presenting the qualitative study.

- 164: How was the mentioned literature in line with the reported study? The author mentioned that the conducted study did not take cultivar into account instead.

- 167 and 230: The reported study did not consider fertilizer as a variable. Why would the author present this discussion?

- 230: The present study did not evaluate the effect of different drying methods.  This discussion was also irrelevant.

- Table 1-4: If the author wishes to put the sample into codes in the table, this should be explained or mentioned beforehand.

- 265: This sentence is not concluding the result and discussion related to the conducted study. It suits more for a metadata study. So, it is better to remove this sentence because the evaluation of 'plant cultivar used, climate, specific agronomic practices, and finally also storage time and conditions' was not conducted in the present study.

- 268: This sentence is not relevant to the study and the manuscript. It is better to remove it from the conclusion.

Author Response

Reviewer 1

Comment: The paper reports the identification of bioactive compounds in several organically and conventionally processed dried fruits. The study is useful in providing information about the beneficial compounds in the studied samples. However, the author is suggested to make some improvements to increase the quality of the manuscript.

Authors’ response: We would like to thank the Reviewer for such a comprehensive review, and for all the very valuable and important suggestions which allowed us to correct errors and significantly improve the quality of the manuscript (as explained in the below replies to the Reviewer’s comments).

Comment : Some irrelevant discussions appear in both the introduction and the discussion sections. The sentence construction and English grammar are also needed considerable help of improvements.

Authors’ response: Following the Reviewer’s suggestion, we have revised the introduction and discussion sections, and removed some sentences covering aspects which were irrelevant for the presented results/topic (see lines 192-199, 251-253, 270-277 in the revised manuscript). We have also introduced a number of language corrections throughout the manuscript.

Comment: The report covered not only qualitative but also quantitative study. So, it would be better to provide both data in the abstract and conclusion.

Authors’ response: As suggested by the Reviewer, we have introduced quantitative data on the concentrations of phenolics and carotenoids into the abstract and the conclusions section of the manuscript (see lines 25-28 and 295-298 in the revised manuscript).

Comment: 24-32: These sentences are not related to the actual study; it is better to remove it from the abstract.

Authors’ response: Following the Reviewer’s suggestion, we have removed the 2 sentences from the abstract (see lines 31-35 in the revised manuscript).

 Comment: Introduction: The storyline is still confusing—no redlining between some sentences nor some paragraphs. The information provided by the author jumps from one description into another. Because the study only used several kinds of fruits, the justification for the selected fruit used in the study is needed. Additionally, a more precise judgment for the chosen analytes is also appreciated.

Authors’ response: We are grateful for this comment, which clearly helped us to improve the Introduction section of the paper. We have now revised thoroughly this section – we significantly improved the storyline of the text. We have added a short justification for the selected fruits (see line 63), and we have kept the short text which justifies the focus on particular groups of analytes in the study (see lines 65-68). We think that now the introduction is much clearer for the readers.

Comment: 38: bioactive compounds is a very broad term. Vitamin, phenolic, and polyphenol are some examples of bioactive compounds. However, phenolics are not vitamin.

Authors’ response: To address the Reviewer’s concerns, and to avoid potential confusion of the reader, the sentence has been revised as follows: “Fruits contain many valuable nutrients and biologically active compounds such as e.g. polyphenols, carotenoids and vitamins, with a positive effect on human health.” (see lines 40-42 in the revised manuscript).

Comment: Materials and methods: It would be better to separate the section for reagents and instruments used in the study. In this way, the reader could follow the information better.

Authors’ response: Following the Reviewer’s suggestion, we have introduced two sub-sections into the Materials and Methods section: ‘2.1. Chemicals’ and ‘2.2. Equipment’ (see lines 90-102 in the revised manuscript).

Comment: Results and discussion: Some irrelevant literature were discussed in this section.

Authors’ response: We have revised the ‘Results and discussion’ section as suggested by the Reviewer. Details have been given in the replies to the below listed comments about specific literature/aspects not corresponding directly to the study results.

Comment: Discussing all the results into one big section is quite confusing. It is better if the author could arrange this section into several sub-sections and discuss the main idea for each topic separately.

Authors’ response: As suggested by the Reviewer, we have arranged the ‘Results and discussion’ section into a number of product-related sub-sections: 3.1. Dried apricots; 3.2. Dried apple rings; 3.3. Dried cranberries; 3.4. Prunes (see lines 157, 207, 226, 254 in the revised manuscript).

Comment: Results and discussion: The identification result needs to be shown before presenting the qualitative study.

Authors’ response: Following the Reviewer’s comment, we have introduced more quantitative results into the ‘Results and discussion’ section (see lines 176-177, 232-233, 257-258 in the revised manuscript).

Comment: 164: How was the mentioned literature in line with the reported study? The author mentioned that the conducted study did not take cultivar into account instead.

Authors’ response: This part of the discussion has been revised to avoid confusion of the readers. It was meant that in the cited study the polyphenols and vitamin A contents in organically grown fruit were found to be higher than in those from conventional cultivation in the case of all tested samples – and this is in line with the present study (see lines 190-195 in the revised manuscript).

Comment: 167: The reported study did not consider fertilizer as a variable. Why would the author present this discussion? 230: The present study did not evaluate the effect of different drying methods.  This discussion was also irrelevant.

Authors’ response: We agree with the Reviewer that this part of discussion was irrelevant, and we have removed it from the “Results and discussion” section of the revised manuscript (see lines 195-199 and 270-277 in the revised manuscript).

Comment: Table 1-4: If the author wishes to put the sample into codes in the table, this should be explained or mentioned beforehand.

Authors’ response: Following the Reviewer’s suggestion, we have introduced information about samples coding into the ‘Materials and methods’ section (see lines 153-154 in the revised manuscript).

Comment: 265: This sentence is not concluding the result and discussion related to the conducted study. It suits more for a metadata study. So, it is better to remove this sentence because the evaluation of 'plant cultivar used, climate, specific agronomic practices, and finally also storage time and conditions' was not conducted in the present study.

Authors’ response: Following the Reviewer’s suggestion, we have removed this sentence from the conclusions section (see lines 307-310 in the revised manuscript).

 Comment: 268: This sentence is not relevant to the study and the manuscript. It is better to remove it from the conclusion.

Authors’ response: Following the Reviewer’s suggestion, we have removed this sentence from the conclusions section (see lines 310-314 in the revised manuscript).

Reviewer 2 Report

The subject of the paper is of interest. English reviewing must be done.

Comments:

Line 14: ….including i.e. higher contents…. Should be…. including i.e. higher contents….

Line 15: …and higher antioxidant…. Should be…and antioxidant….

Line 24: …..between… should be….among….

Line 25:….in case…should be…..in the case……

Line 27: …products, there was either no consistent significant…..

Line 29:….organic fruit and vegetables…..

Line 43:…..reduce the level of LDL…..

Line 48:…..and thus, the availability……

Line 51:….kiwifruit….

Line 73:…in the case of…

Line 86: What do you mean by a scale method?

Line 89: …..content in the examined…..

Line 90: Be consistent through the text, use either g.g-1 or g/g

Line 111:….HPLC (Shimadzu…..

Line 134: (varieties not specified). There was known the production time? Storage time could have effect too and shall be mentioned.

Line 133: results among….due to different drying methods,…..

Line 144: Reference Salur-Can et al. is not numbered and does not appear in the reference list.

Line 147: Replace polyphenols (sum) by total polyphenols, this shall be for all components through the text and tables.

Line 152: Take out the year through the text when you write the names as Baranskiet al. (16)

Line 164: …in the present study.

Line 173: They concluded that the industrial processing…….

Line 178: If we compare 2 items we write between, if we compare more than 2 items we write among. Correct that through the text. It is among brands and between the 2 production systems.

Line 182: The effects of the production system were identified only in the……

Lines 185 and 186: replace between by among

Line: 201: I do not understand what you mean….benzoic acid and app being….

Do you mean …benzoic acid application being widely…?

Line 214: …., compared to the initial…

Lines 216-217: This last phrase is incorrect. …Did not exceed 50ºC? High temperatures decrease the molds, 50ºC is not the limit.

Line 218: In the case of prunes……among the brands…..

Line 224: This compound does not appear naturally in plum fruit. Provide a reference for this statement.

Line 227: … the effect of….

Line 234: Dryed under solar..

Line 235: contents, as compared…..

In the tables the units provided in the bottom of the table should be under the compound name. Also there is no need to number the means. A note should be for the statistics reference (number 5 to be replaced by 1 ). Clarify which level of significance you have in the bottom of the table,  so there is no need for the p-value Brand column, only for the production system.

Also replace sum by total carotenoids, etc.

Line 254: This study confirms….

Line 255: …polyphenols. However,….

Line 256:…..among…

Line 259: In the case of the other products…

Line 263: ….superior quality…..

Line 265:…such as plant……

Line 266: …..and storage time…..

Line 267:…..products’ quality…..

Author Response

Reviewer 2

Comment: The subject of the paper is of interest. English reviewing must be done.

Authors’ response: We would like to thank the Reviewer for the comprehensive review and for all the valuable comments and suggestions. We have revised the manuscript following each of the comments (as explained below), which clearly improved the quality of the revised manuscript.

 Comment: Line 14: ….including i.e. higher contents…. Should be…. including i.e. higher content….

Authors’ response: We have revised the sentence as requested (see line 14 in the revised manuscript).

 Comment: Line 15: …and higher antioxidant…. Should be…and antioxidant….

Authors’ response: We have deleted the word “higher” from the sentence, as requested (see line 15 in the revised manuscript).

 Comment: Line 24: …..between… should be….among….

Authors’ response: We have changed ‘between’ to ‘among’, as suggested (see line 24 in the revised manuscript).

 Comment: Line 25:….in case…should be…..in the case……

Authors’ response: We have changed ‘in case’ to ‘in the case’, as suggested (see line 28 in the revised manuscript).

 Comment: Line 27: …products, there was either no consistent significant…..

Authors’ response: We have removed comma from the sentence, as suggested (see line 30 in the revised manuscript).

Comment: Line 29:….organic fruit and vegetables…..

Authors’ response: This whole sentence has been removed from the manuscript, following a suggestion of Reviewer 1. Therefore, this correction has not been introduced (see lines 31-35 in the revised manuscript).

Comment: Line 43:…..reduce the level of LDL…..

Authors’ response: We have added “the” to the sentence, as suggested (see line 46 in the revised manuscript).

Comment: Line 48:…..and thus, the availability……

Authors’ response: We have revised the sentence following the Reviewer’s request (see line 51 in the revised manuscript).

Comment: Line 51:….kiwifruit….

Authors’ response: We have changed ‘kiwi’ to ‘kiwifruit’, as suggested (see line 54 in the revised manuscript).

Comment: Line 73:…in the case of…

Authors’ response: This whole sentence has been revised, following a suggestion of Reviewer 1. Therefore, this correction has not been introduced (see line 85 in the revised manuscript).

 Comment: Line 86: What do you mean by a scale method?

Authors’ response: We meant a gravimetric method. This wording (name of the method) has now been revised  (see line 111 in the revised manuscript).

Comment: Line 89: …..content in the examined…..

Authors’ response: We have corrected this error (we have introduced “in” to the sentence) (see line 115 in the revised manuscript).

Comment: Line 90: Be consistent through the text, use either g.g-1 or g/g

Authors’ response: We have now unified this throughout the text (and used g 100 g-1 style) (see lines 25, 26, 27, 28, 115, 160, 177, 210, 233, 246, 258, 279, 282, 285, 288, 296, 297, 298 in the revised manuscript).

Comment: Line 111:….HPLC (Shimadzu…..

Authors’ response: We have moved the Shimadzu equipment description to a new subsection (“2.2. Equipment”) which was developed following the suggestion of Reviewer 1. We have removed the word “device” from the description, following the Reviewer’s suggestion (see line 100 in the revised manuscript).

Comment: Line 134: (varieties not specified). There was known the production time? Storage time could have effect too and shall be mentioned.

Authors’ response: In the revised manuscript we have also mentioned storage time as one of the factors that could have impacted on the concentrations of the measured compounds in the samples (see line 163 in the revised manuscript). However, we had no information about the exact production time (and storage time) of the samples.

Comment: Line 133: results among….due to different drying methods,…..

Authors’ response: We have changed ‘between’ to ‘among’ and removed the word “technical” from the sentence, as suggested (see line 161 and 162 in the revised manuscript).

Comment: Line 144: Reference Salur-Can et al. is not numbered and does not appear in the reference list.

Authors’ response: We would like to thank the Reviewer for pointing this. We have now corrected the citation in the text (see line 173 in the revised manuscript) and added this reference to the reference list (see lines 382-383 in the revised manuscript).

 Comment: Line 147: Replace polyphenols (sum) by total polyphenols, this shall be for all components through the text and tables.

Authors’ response: As suggested by the Reviewer, we have replaced “polyphenols (sum)” by “polyphenols (total)” throughout the manuscript. Similar corrections have been introduced also for other groups of compounds (see lines 176, 178, 209, 215, 232, 257, 260, all Tables,  in the revised manuscript).

Comment: Line 152: Take out the year through the text when you write the names as Baranski et al. (16).

Authors’ response: Following the Reviewer’s suggestion, we have taken out the year from the text in all cases when we write the names of the authors of the cited publications (see lines 129, 147, 173, 182, 189, 201, 239, 241, 246, 266 in the revised manuscript).

Comment: Line 164: …in the present study.

Authors’ response: As suggested by the Reviewer, we have replaced “presented” by “present” (see line 195 in the revised manuscript).

 Comment: Line 173: They concluded that the industrial processing…….

Authors’ response: We have corrected the sentence following the Reviewer’s suggestion (we have replaced “it was concluded” by “they concluded”) (see line 203 in the revised manuscript).

Comment: Line 178: If we compare 2 items we write between, if we compare more than 2 items we write among. Correct that through the text. It is among brands and between the 2 production systems.

Authors’ response: As advised by the Reviewer, we have corrected among/between throughout the text (see lines 24, 161, 210, 218, 219, 230, 255, 295 in the revised manuscript).

Comment: Line 182: The effects of the production system were identified only in the……

Authors’ response: The sentence was corrected as suggested (see line 214 in the revised manuscript).

Comment: Lines 185 and 186: replace between by among

Authors’ response: As already mentioned, we have corrected among/between throughout the text, following the Reviewer’s comment. These two cases have also been corrected (see lines 218 and 219 in the revised manuscript).

 Comment: Line: 201: I do not understand what you mean….benzoic acid and app being…. Do you mean …benzoic acid application being widely…?

Authors’ response: It was supposed to be written: “benzoic acid and sodium benzoate being widely used…”. Part of the sentence was accidently deleted during the manuscript preparation. It has now been corrected (see line 236 in the revised manuscript).

Comment: Line 214: …., compared to the initial…

Authors’ response: Following the Reviewer’s suggestion, we have replaced “with” with “to” in this sentence (see line 250 in the revised manuscript).

Comment: Lines 216-217: This last phrase is incorrect. …Did not exceed 50ºC? High temperatures decrease the molds, 50ºC is not the limit.

Authors’ response: This whole sentence has been removed from the manuscript, following a suggestion of Reviewer 1. Therefore, this correction has not been introduced (see line 251-253 in the revised manuscript).

Comment: Line 218: In the case of prunes……among the brands…..

Authors’ response: We have corrected the sentence following the Reviewer’s suggestion (see line 255 in the revised manuscript).

Comment: Line 224: This compound does not appear naturally in plum fruit. Provide a reference for this statement.

Authors’ response: This statement was not confirmed by the scientific literature, and therefore has been removed from the discussion in the revised manuscript (see line 262-263 in the revised manuscript).

Comment: Line 227: … the effect of….

Authors’ response: We have changed “the effects of” to “the effect of”, as requested by the Reviewer (see line 266 in the revised manuscript).

Comment: Line 234: Dryed under solar..

Authors’ response: This whole sentence has been removed from the manuscript, following a suggestion of Reviewer 1. Therefore, this correction has not been introduced (see lines 270-277 in the revised manuscript).

Comment: Line 235: contents, as compared…..

Authors’ response: This whole sentence has been removed from the manuscript, following a suggestion of Reviewer 1. Therefore, this correction has not been introduced (see lines 270-277 in the revised manuscript).

Comment: In the tables the units provided in the bottom of the table should be under the compound name. Also there is no need to number the means. A note should be for the statistics reference (number 5 to be replaced by 1 ). Clarify which level of significance you have in the bottom of the table,  so there is no need for the p-value Brand column, only for the production system. Also replace sum by total carotenoids, etc.

Authors’ response: Following the Reviewer’s suggestion, we have placed the units next to the names of the groups of chemical compounds in all 4 tables. We have also removed the numbers and footnotes about the means and SD. As suggested, the level of significance has been clarified in the footnote of the tables, and the brand p-value column has been removed. All “sums” have been replaced by “totals” (see Tables 1-4 in the revised manuscript).

Comment: Line 254: This study confirms….

Authors’ response: As requested by the Reviewer, the statement “The study confirmed” was replaced by “This study confirms” (see line 293 in the revised manuscript).

Comment: Line 255: …polyphenols. However,….

Authors’ response: The sentence has been corrected as requested (divided into 2 sentences) (see line 294 in the revised manuscript).

Comment: Line 256:…..among…

Authors’ response: We have changed ‘between’ to ‘among’, as suggested (see line 295 in the revised manuscript).

Comment: Line 259: In the case of the other products…

Authors’ response: We have added “the”, as requested (see line 301 in the revised manuscript).

Comment: Line 263: ….superior quality…..

Authors’ response: As requested, “some superior qualities” was changed to “superior quality” (see line 304 in the revised manuscript).

Comment: Line 265:…such as plant……

Authors’ response: This whole sentence has been removed from the manuscript, following a suggestion of Reviewer 1. Therefore, this correction has not been introduced (see lines 307-310 in the revised manuscript).

 Comment: Line 266: …..and storage time…..

Authors’ response: As mentioned above, this whole sentence has been removed from the manuscript, following a suggestion of Reviewer 1. Therefore, this correction has not been introduced (see line 307-310 in the revised manuscript).

Comment: Line 267:…..products’ quality…..

Authors’ response: As mentioned above, this whole sentence has been removed from the manuscript, following a suggestion of Reviewer 1. Therefore, this correction has not been introduced (see line 307-310 in the revised manuscript).

Reviewer 3 Report

From my point of view, this type of comparative studies on fruits is very interesting. But I think that the approach is not correct, since fruits should be selected in which the same parameters could be compared. The result obtained with apricots regarding the increase of carotenoids in organic cultivation is interesting. It would be interesting if it could be compared with other fruits that contain carotenoids. In comparative studies the parameters to be compared must be the same. For be scientifically relevant, it is necessary to add other fruits to determine the carotene content.
I also believe that more information is needed about cultivation conditions, soil, etc.

Author Response

Reviewer 3

Comment: From my point of view, this type of comparative studies on fruits is very interesting. But I think that the approach is not correct, since fruits should be selected in which the same parameters could be compared. The result obtained with apricots regarding the increase of carotenoids in organic cultivation is interesting. It would be interesting if it could be compared with other fruits that contain carotenoids. In comparative studies the parameters to be compared must be the same. For be scientifically relevant, it is necessary to add other fruits to determine the carotene content. I also believe that more information is needed about cultivation conditions, soil, etc.

Authors’ response: We would like to thank the Reviewer for the valuable comments and suggestion for the research approach, where carotenoids profiles of different fruits could be compared. We will surely consider such a focus in our future studies. However, we would like to underline that in this study the main focus was on the comparison between organic vs. conventional product, and not the comparison between different products (apricots, apple rings, cranberries and prunes). The compared parameters were the same, i.e.: (a) carotenoids in organic vs. conventional apricots or (b) particular phenolics in organic vs. conventional cranberries etc. We wanted to investigate whether the production system effect (organic vs. conventional) observed in the quality of crops in many well-controlled agronomic trials can be also seen in the processed (dried) products available on the market.

We agree that to explain the background of differences (or lack of differences) between the analysed organic vs. conventional products we should have information about the cultivation conditions, soil, plant variety, and many other agronomic and environmental factors. However, in this study (a “basket study”) we wanted to point researchers’ and consumers’ attention to the fact that: even if organic fruits and vegetables are often found to be richer in many bioactive compounds than conventionally produced ones, it does not necessarily mean that it’s true for each random product present on the market, and that they carry these health-promoting qualities through the processing step.

Thus, actions should be taken to promote practices which could help to assure health-promoting attributes of organic fruit and vegetables, and to bring these health-promoting attributes from field to fork through the processing step, to reach the consumers’ expectations. We hope that the revised manuscript adequately addresses/points attention to the above mentioned aspects.

Round 2

Reviewer 1 Report

The authors have revised and improved the quality of the paper. However, there is still a point of revision that has not done related to the following info:

Comment: Table 1-4: If the author wishes to put the sample into codes in the table, this should be explained or mentioned beforehand.

Authors’ response: Following the Reviewer’s suggestion, we have introduced information about samples coding into the ‘Materials and methods’ section (see lines 153-154 in the revised manuscript).

The revised part is not available in the mentioned lines in the ms 

Author Response

Reviewer 1

Comment: The authors have revised and improved the quality of the paper. However, there is still a point of revision that has not done related to the following info: Comment: Table 1-4: If the author wishes to put the sample into codes in the table, this should be explained or mentioned beforehand. Authors’ response: “Following the Reviewer’s suggestion, we have introduced information about samples coding into the ‘Materials and methods’ section (see lines 153-154 in the revised manuscript).” The revised part is not available in the mentioned lines in the ms.

Authors’ response: We are very grateful for the Reviewer’s positive feedback on our revised manuscript. To address the above mentioned Reviewer’s comment more adequately, we have moved the text about the brands’ coding to the ‘2.3. Samples’ subsection of the ‘Materials and methods’ section of the manuscript, and extended the text as follows: “For each of the 4 tested products, brand names were coded with numbers (1-6). These number codes were consequently used in the results’ tables.” (see lines 106-108 in the revised manuscript).

Reviewer 3 Report

The work has improved. I understand the authors' point of view when doing a more informative work, aimed at the consumer. But from the scientific point of view, for the conclusions to be more relevant, it is necessary that for comparison between organic and conventional products, direct agronomic aspects must be controlled.

Author Response

Reviewer 3

Comment: The work has improved. I understand the authors' point of view when doing a more informative work, aimed at the consumer. But from the scientific point of view, for the conclusions to be more relevant, it is necessary that for comparison between organic and conventional products, direct agronomic aspects must be controlled.

Authors’ response: We would like to thank the Reviewer for the positive feedback on our revised manuscript. We agree of course that it would be of great interest and scientific relevance to have access to the extensive set of information about agronomic, environmental and processing-related moderators of the quality attributes of the products available on the market – this would allow to investigate and explain where the specific products’ quality attributes come from, and would surely allow to draw more complex conclusions. At the same time so called ‘basket study’ approach undertaken here, giving the consumers and scientists an indication of what’s available on the market, is also widely practiced – we provide below a few examples of such “basket study” research papers:

  1. Wei, D. Wang, Y., Jiang, D., Feng, X., Li, X., Wang M. Survey of Alternaria Toxins and Other Mycotoxins in Dried Fruits in China, Toxins, 2017, 9, 200; doi:10.3390/toxins9070200 - A total of 220 samples consisting of raisins (57), dried apricots (56), dried dates (53) and dried, wolfberries (54) were randomly collected from different supermarkets and local markets in Beijing, China, during 2016~2017.
  2. Wong, R., Kim, S., Chung S-J., Cho M-S. Texture Preferences of Chinese, Korean and US Consumers: A Case Study with Apple and Pear Dried Fruits, Foods, 9, 377; doi:10.3390/foods9030377 - Fourteen types of fruit chips (seven pear and seven apple chips) with four textural properties (crispy, jelly-like, puy, and soft) were selected as the products of interests in the hedonic mapping experiments. All of the dried fruit chips from Korea, China, and the US were purchased online.
  3. Proestos Ch.,Komaitis M. (2013). Analysis of naturally occurring phenolic compounds in aromatic plants by RP-HPLC coupled to Diode Array Detector (DAD) and GC-MS after silylation, Foods, 2, 90-99; doi:10.3390/foods2010090 - Samples of Origanum dictamnus (dictamnus), Eucalyptus globulus (eucalyptus), Origanum vulgare (oregano), Mellisa officinalis L. (balm mint) and Sideritis cretica (mountain tea) were obtained from local stores.
  1. Majdi Ch., Pereira C., Dias M.I., Calhelha R.C., Alves M.J., Rhourri-Frih B., Charrouf Z., Barros L., Amaral J.S., Ferreira I.C.F.R. (2020). Phytochemical Characterization and Bioactive Properties of Cinnamon Basil (Ocimum basilicum Cinnamon’) and Lemon Basil (Ocimum x citriodorum), Antioxidants 2020, 9, 369; doi:10.3390/antiox9050369 - Commercial samples (bags of 50 g) of fragmented dried leaves of “cinnamon basil” (O. basilicum cv. ’Cinnamon’) and of “lemon basil” (Ocimum x citriodorum), were provided by a Portuguese company (Cantinho das Aromáticas, Vila Nova de Gaia, Portugal) dedicated to the production and commercialization of dry aromatic herbs, produced in the Northern region of Portugal under organic farming.